# Effect of Antimicrobial Photodynamic Therapy on the Tongue Dorsum on Reducing Halitosis and the Duration of the Effect: A Randomized Clinical Trial

**DOI:** 10.3390/healthcare12100980

**Published:** 2024-05-09

**Authors:** Takayuki Maruyama, Daisuke Ekuni, Aya Yokoi, Junichiro Nagasaki, Nanami Sawada, Manabu Morita

**Affiliations:** 1Department of Preventive Dentistry, Faculty of Medicine, Dentistry and Pharmaceutical Sciences, Okayama University, Okayama 700-8558, Japan; dekuni7@md.okayama-u.ac.jp (D.E.); yokoi-a1@cc.okayama-u.ac.jp (A.Y.); 2Advanced Research Center for Oral and Craniofacial Sciences, Okayama University Dental School, Okayama 700-8558, Japan; 3Okayama University Dental School, Okayama 700-8558, Japan; p33x9th0@s.okayama-u.ac.jp; 4Department of Preventive Dentistry, Okayama University Hospital, Okayama 700-8558, Japan; de422027@s.okayama-u.ac.jp; 5Department of Oral Health Sciences, Takarazuka University of Medical and Health Care, Takarazuka 666-0162, Japan; m.morita@tumh.ac.jp

**Keywords:** halitosis, antimicrobial photodynamic therapy, prevention, randomized clinical trial

## Abstract

Antimicrobial photodynamic therapy (PDT) is a treatment that is gaining popularity in modern clinical medicine. However, little is known about the effect of PDT alone on reducing oral halitosis and the duration of the effect. This trial examined the effect of PDT on the tongue dorsum on reducing oral halitosis and the duration of the effect. This study was approved by the Ethics Committee of Okayama University Graduate School of Medicine, Dentistry, and Pharmaceutical Sciences, and Okayama University Hospital (CRB20-015), and it was registered in the Japan Registry of Clinical Trials (jRCTs061200060). Twenty-two participants were randomly assigned to two groups: an intervention group and control group. PDT was performed in the intervention group using red laser emission and methylene blue gel on the middle and posterior area of the tongue dorsum. The concentration of volatile sulfur compounds, bacterial count on the tongue dorsum, probing pocket depth, bleeding on probing, and simplified oral debris index score were determined before and 1 week after PDT. The Mann–Whitney *U* test was used to assess the significance of the differences in each parameter between the two groups. We found that the hydrogen sulfide concentration and bacterial count on the tongue dorsum were decreased in the intervention group, but there was no statistically significant difference between the two groups. These results indicated that performing only PDT on the tongue dorsum may not contribute to reducing halitosis.

## 1. Introduction

Halitosis is an unpleasant odor in the oral cavity that has a significant impact on social interactions and mental health in humans [1]. During the coronavirus disease 2019 pandemic, wearing a face mask was strongly recommended. As a result, people have become increasingly aware of halitosis in the confined space inside a mask. The prevalence of halitosis is estimated to be 31.8% [2], and in most cases, halitosis originates in the oral cavity [3]. Halitosis is caused by the presence of volatile sulfur compounds (VSCs), which are produced by bacteria during the metabolism of proteinaceous substrates in the oral cavity. The main substances in VSCs are hydrogen sulfide, methyl mercaptan, and dimethyl sulfide [4]. The tongue is thought to be the main source of oral halitosis due to its large surface area. In addition, the irregular morphology of the papillary structure of the tongue dorsum promotes bacterial attachment and the accumulation of organic substances, such as necrotic epithelial cells, food debris, and blood, which form the tongue coating [5]. The resulting tongue coating is a major cause of VSCs [6].

Traditional treatments for oral halitosis have focused on mechanical and chemical methods of eliminating oral bacteria [7,8]. Oral bacteria can be mechanically removed by tooth brushing and cleaning of the tongue with a scraper. In addition, toothpaste or mouthwash containing chlorhexidine and antimicrobials can be used to chemically remove oral bacteria or mask odors [9]. However, these methods are not very effective and/or the effects are not long-lasting. A meta-analysis showed that there was very low evidence for results in dentist-reported organoleptic testing scores for mechanical tongue cleaning with a tongue scraper vs. no tongue cleaning, using 0.3% triclosan toothpaste vs. a control toothpaste, and using a mouthwash containing chlorhexidine and zinc acetate vs. a placebo mouthwash. Therefore, new approaches for reducing oral halitosis are required.

Antimicrobial photodynamic therapy (PDT) is a treatment that is gaining popularity in modern clinical medicine. In PDT, a photosensitizer is injected into the infected area and irradiated with light of a specific wavelength to generate reactive oxygen, which destroys the cell wall of bacteria and causes cell death [10]. PDT has no side effects except hypersensitivity to the photosensitizer. Also, PDT does not lead to the development of antibiotic-resistant bacteria, it helps maintain the oral microbiota, and it has low toxicity [11]. In dentistry, PDT has been applied to periodontal treatment [12] and endodontics [13]. A systematic review of periodontal therapy evaluated studies using PDT as an adjunct to scaling and root planing (SRP) with a 3-month follow-up. The results showed that the combination of SRP and PDT significantly reduced the periodontal pocket depth (PPD) compared to SRP alone [12]. Also, a systematic review of endodontics showed that PDT significantly eliminated bacteria even against antibiotic-resistant species and was also effective in disrupting biofilms in laboratory studies [13]. On the other hand, the combined effect of mechanical tongue cleaning and PDT on reducing oral halitosis has been examined in previous studies [14,15]. However, little is known about the effect of using PDT alone on oral halitosis and the duration of the effect. The null hypothesis in this study was that there would be no effect of PDT alone on oral halitosis. Thus, the purpose of this study was to examine the effect of PDT on the tongue dorsum on reducing halitosis and the duration of the effect.

## 2. Materials and Methods

### 2.1. Trial Design

This study was a single-masked, parallel, single-center, randomized clinical trial that followed the Declaration of Helsinki and the Consolidated Standards of Reporting Trials (CONSORT) guidelines and checklist. The study was approved by the Ethics Committee of Okayama University Graduate School of Medicine, Dentistry, and Pharmaceutical Sciences, and Okayama University Hospital (CRB20-015), and it was registered in the Japan Registry of Clinical Trials (jRCTs061200060).

### 2.2. Blinding

The periodontal examiner, the investigator who measured the VSC concentrations, and the investigator responsible for data analysis were blinded to the treatment assignment.

### 2.3. Participants

Participants were recruited from among the students at Okayama University Dental School. Those who complained of oral halitosis and were over 20 years of age were included. Those with oral cancer, pregnancy, breastfeeding, or hypersensitivity to the photosensitizer were excluded. All participants received an explanation of the objectives and procedures of this study, and they all provided written informed consent for participation. Periodontal treatment such as SRP, tongue cleaning with a tongue scraper, and the use of an oral moisturizer or gargle were prohibited during the study period as they could have affected the outcomes of the study.

### 2.4. Randomization

Each participant was given a code number, and the coordinator (N.S.) used a computer-generated table to randomly assign the participants to one of two groups, i.e., the intervention group or the control group. The allocation ratio was 1:1.

### 2.5. Sample Size Calculation

The sample size was estimated based on a previous study [16], assuming a mean difference in hydrogen sulfide concentration of 3.88 ng/10 mL and a standard deviation of 3.40 ng/10 mL between the intervention group and the control group at 1 week after treatment. Based on the data, it was determined that 11 participants per group were required to provide an alpha value of 0.05 and 80% power for two-tailed and unpaired *t*-tests.

### 2.6. Intervention

PDT was performed using a laser-activated disinfection device (FotoSan630, CMS Dental Inc., Roslev, Denmark) with red laser emission (wavelength: 630 nm, irradiance: 2000–4000 mW/cm^2^, spot size at the probe tip: diameter of 8 mm). High-viscosity methylene blue gel (0.1 mg/mL) (FotoSan Agent, CMS Dental Inc.) was applied at 1 cm intervals for a total of six spots on the center left, the center midline, the center right, the posterior left, the posterior midline, and the posterior right side of the tongue dorsum. Then, each spot was irradiated for 30 s. During the laser irradiation, the participants and operator wore goggles to protect their eyes from the laser. After PDT, the methylene blue gel was completely washed away with sterile purified water.

In the control group, there was no treatment.

### 2.7. Outcome Assessment

The concentration of VSCs, bacterial counts on the tongue dorsum, probing pocket depth (PPD), bleeding on probing (BOP), and simplified oral debris index (DI-S) score were determined before and 1 week after PDT (Figure 1). Each measurement was performed at the same time in the morning (between 7 and 9 AM).

The concentrations of VSCs (hydrogen sulfide, methyl mercaptan, and dimethyl sulfide) were measured using a portable gas chromatography device (OralChroma, Nissha FIS Inc., Osaka, Japan). The participants were instructed not to consume garlic, onions, and alcohol and to use antiseptic mouthwashes 24 h prior to the VSC measurements. They were also instructed not to eat breakfast, brush their teeth, smoke, and use perfume on the day of the VSC measurements and to drink water or gargle 1 h prior to the VSC measurements. Air from the oral cavity was collected according to the manufacturer’s instruction. In brief, each participant was instructed to keep their mouth closed for 30 s, and a syringe was inserted into the mouth to collect 1 mL of air, which was injected into the portable gas chromatography device using a needle [17].

A portable bacteria counter (Saikin Counter, Panasonic Healthcare Co., Ltd., Tokyo, Japan) was used to measure the bacterial counts on the tongue dorsum. This device was based on the dielectrophoretic impedance measurement (DEPIM) method. DEPIM is a method of measuring the bacterial counts in a liquid sample on electrodes using dielectrophoretic force, measuring the change in impedance between the electrodes, and converting to the concentration of bacteria in the sample [18]. A sample of the tongue coating was collected from the middle of the tongue dorsum by gently wiping the tongue surface with a cotton swab according to the manufacturer’s instructions [18]. The bacterial counts are expressed as colony-forming units (CFU) per milliliter.

One dentist (A.Y.) evaluated the PPD, BOP, and DI-S. The PPD was measured at six sites (mesio-buccal, mid-buccal, disto-buccal, mesio-lingual, mid-lingual, and disto-lingual) on all teeth except for the third molars using a color-coded periodontal probe (CP-11 Color-Coded Probe, Hu-Friedy, Chicago, IL, USA) [19]. BOP was assessed by the presence or absence of bleeding from periodontal pockets on probing measurements, and a score was given to each individual tooth. Then, the percentage of sites with BOP was calculated. The DI-S was scored for six representative teeth such as the maxillary right central incisor (11), maxillary right (16) and left (26) first molars, mandibular left central incisor (31), and mandibular left (36) and right (46) first molars [20]. The DI-S was scored from 0 to 3 from the debris accumulated on the surface of each tooth: code 0 = no debris on the tooth surface; code 1 = debris presence covering less than 1/3 of the tooth surface; code 2 = debris presence covering more than 1/3 of the tooth surface; code 3 = debris presence covering more than 2/3 of the tooth surface [20]. Additionally, the PPD was measured again 2 weeks later in two volunteers. The intra-examiner reliability of the PPD measurements as evaluated by kappa statistics was >0.8. All variables that we assessed in this study and brief information about the outcome assessment are shown in Table 1.

### 2.8. Statistical Analysis

The rate of change between baseline and 1 week after PDT for each subject was calculated for the concentration of VSCs and bacterial counts on the tongue dorsum. The formula for calculating the rate of change is as follows: (1 week after PDT—baseline)/baseline × 100. The Mann–Whitney *U* test was used to assess the significance of the differences for each parameter between the two groups. All analyses were performed using SPSS 27.0J for Windows (IBM Japan, Tokyo, Japan). Values of *p* < 0.05 were considered to indicate statistical significance.

## 3. Results

Figure 2 shows the protocol of this study. No participants dropped out during the study period. Also, no participants changed their behaviors of oral hygiene during the study period.

Table 2 shows the characteristics of the participants at baseline. The median hydrogen sulfide concentration in the intervention group was 188.0 ppb. Defining halitosis as a concentration of hydrogen sulfide above 112 ppb [17], seven subjects in the intervention group and five subjects in the control group were diagnosed as having halitosis. There were no significant differences in all the valuables between the groups (*p* > 0.05, Mann–Whitney’s *U* test).

Table 3 shows the characteristics of the participants at 1 week after PDT. The bacterial counts on the tongue dorsum in the intervention group were significantly higher than those in the control group. On the other hand, there were no significant differences in the other parameters between the intervention group and the control group.

Table 4 shows the rate of change between baseline and 1 week after PDT for each parameter in the two groups. In the intervention group, the concentrations of hydrogen sulfide, methyl mercaptan, and dimethyl sulfide and the bacterial counts on the tongue dorsum were decreased. On the other hand, in the control group, the concentrations of hydrogen sulfide and dimethyl sulfide and the bacterial counts on the tongue dorsum were decreased; however, the concentration of methyl mercaptan was increased. There was no statistically significant difference between the two groups.

## 4. Discussion

To the best of our knowledge, this is the first study to examine the effect of performing only PDT on the tongue dorsum on reducing halitosis and the duration of the effect. The null hypothesis was that there would be no effect of performing PDT alone on oral halitosis. One week after PDT, the hydrogen sulfide concentration and bacterial count on the tongue dorsum were decreased in the intervention group, but there was no statistically significant difference between the groups. It has been reported that penetration of the photosensitizer between the papillae of the tongue affects the thickness of the intraoral biofilm [14]. In addition, the rough surface of the tongue dorsum due to papillae or fissures lowers the oxygen concentration, which promotes the recolonization of anaerobic oral bacteria [21]. Also, singlet oxygen, which is the main effector of PDT, has a short half-life and cannot exert long-term antibacterial effects [22]. Therefore, it is possible that recolonization occurred from the bacteria in the deep papillae of the tongue dorsum, which were out of reach of the laser or outside of the PDT area, and the microbiota thus returned to baseline levels after 1 week. Increasing the number of PDT sessions may be effective against halitosis.

The concentrations of methyl mercaptan and dimethyl sulfide were not significantly decreased in the intervention group. Methyl mercaptan is found mainly in periodontal pockets, and dimethyl sulfide is found mainly in the gastrointestinal tract or internal organs [17]. In this study, since PDT was performed only on the tongue dorsum and not on the periodontal pocket or internal organs, it likely had no effect on the bacteria that produce methyl mercaptan or dimethyl sulfide, and thus, no significant change was seen after 1 week.

In this study, we measured the bacterial counts on the tongue dorsum; however, we did not analyze the oral microbiota. VSCs are mainly produced by Gram-negative anaerobic bacteria, such as *Porphyromonas gingivalis*, *Prevotella intermedia*, *Treponema denticola*, and *Tannerella forsythensis* [15]. A previous study showed that a combination of PDT and tongue cleaning with a tongue scraper could remove more *P. gingivalis* than a tongue scraper alone [22]. In the present study, the bacterial counts on the tongue dorsum in the intervention group were decreased 1 week after PDT; however, no significant difference was observed when compared to the control group. Although PDT reduced the amount of bacteria involved in VSC production, it may not have been effective enough to significantly reduce the amount of bacteria that are not involved in VSC production on the tongue dorsum.

In the present study, the hydrogen sulfide concentration and bacterial count on the tongue dorsum were decreased 1 week after PDT, but there was no statistically significant difference between the groups. Some studies have reported that PDT reduced the concentration of VSCs and that the effect persisted even after 1 week [23,24]. In the study by Romero et al. [23], the PDT group showed a reduction in hydrogen sulfide concentrations even after 1 week, with the mean values remaining 3 times lower than in the control group. Also, in the study by Llanos do Vale et al. [24], the mean concentration of hydrogen sulfide was lower in the PDT group (39 ppb) than in the control group (218.2 ppb). In contrast, another study reported that no persistent effect was observed [25]. Comparing the median hydrogen sulfide concentrations, the reduction rate was 97.6% in the PDT group immediately after treatment. However, no statistically significant differences were found between the PDT and the control groups when comparing before and 1 week after treatment. This discrepancy might be due to various factors. First, the oral environment, e.g., the presence or absence of dental caries, periodontal disease, and dry mouth, may have differed between the studies. Also, the age of the study participants differed. Moreover, some participants in one study had removable complete dentures [24]. These differences in the oral environment may have influenced the composition of the oral microbiota, which may have affected the results of the studies. Second, differences in the PDT protocols, such as the laser type, wavelength, power, and irradiation time, may have affected the results. In this study, the laser wavelength was 630 nm, laser irradiance was 2000–4000 mW/cm^2^, and irradiation time was 30 s per location. On the other hand, in the study by Romero et al. [23], the laser wavelength was 660 nm, laser irradiance was 3537 mW/cm^2^, and irradiation time was 60 s per location. Also, in the study by Llanos do Vale et al. [24], the laser wavelength was 660 nm, laser irradiance was 35,368 mW/cm^2^, and irradiation time was 90 s per location. The decrease in VSC concentration may not have occurred in the present study because the laser intensity and laser irradiation time were different from other studies.

There were some limitations in this study. First, the participants in the present study were fairly young, and different results may be obtained in middle-aged and elderly participants. Second, we only measured the bacterial counts on the tongue dorsum and not in the whole oral cavity. Although the oral hygiene of the participants was fairly good, it is possible that dental plaque may have affected the results. Also, DI-S is one of the indicators for evaluating the oral hygiene status, and six representative teeth were evaluated. Therefore, it may not reflect the total amount of dental plaque in the oral cavity. Third, all the participants were students at Okayama University Dental School, and the generalizability of the results may therefore be limited. Fourth, we conducted one-time PDT session. Other conditions will be required in future studies to identify the effects of PDT on halitosis. Finally, although the sample size calculation was performed, the sample size was small, with 11 participants in each group. Large-scale clinical trials will need to be conducted to validate the results of this study.

## 5. Conclusions

The hydrogen sulfide concentration and bacterial count on the tongue dorsum were decreased by PDT in the intervention group, but there was no statistically significant difference between the groups.

## Figures and Tables

**Figure 1 healthcare-12-00980-f001:**
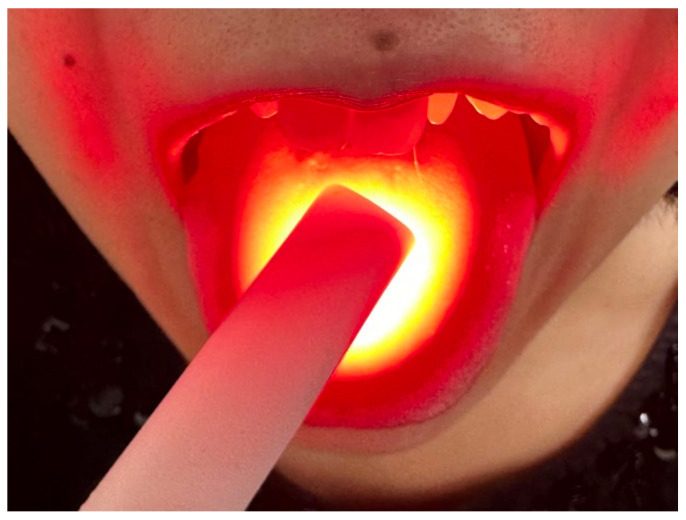
PDT on the tongue dorsum.

**Figure 2 healthcare-12-00980-f002:**
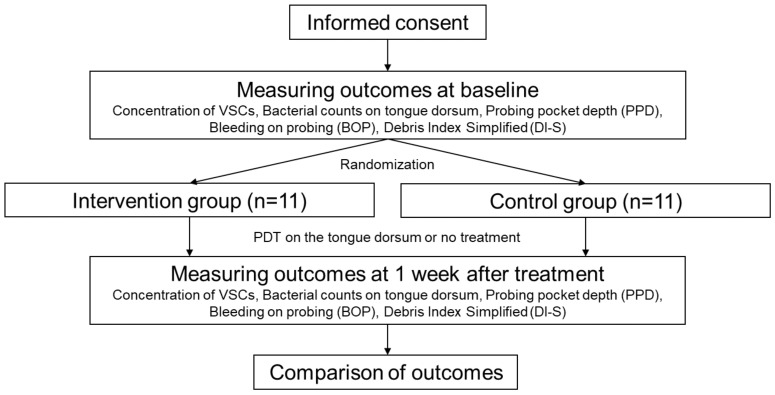
The protocol of this study.

**Table 1 healthcare-12-00980-t001:** Variables and brief information in the outcome assessment.

Variable	Brief Information
Hydrogen sulfide	A main compound causing physiologic halitosis
Methyl mercaptan	A main compound causing halitosis associated with periodontal disease
Dimethyl sulfide	A main compound contributing to extra-oral or blood-borne malodor
Probing pocket depth (PPD)	An index referring the status of periodontal disease
Bleeding on probing (BOP)	An index referring to the activity of periodontal disease
Simplified oral debris index (DI-S)	An index referring to the oral hygiene status

**Table 2 healthcare-12-00980-t002:** Characteristics of participants at baseline.

	Intervention Group (N = 11)	Control Group (N = 11)
Age (years)	30.0 (22.0, 33.0) ^1^	23.0 (21.0, 29.0)
Male	5 (45.5) ^2^	5 (45.5)
Concentration of hydrogen sulfide (ppb)	188.0 (23.0, 666.0)	48.0 (30.0, 347.0)
Concentration of methyl mercaptan (ppb)	92.0 (35.0, 248.0)	25.0 (12.0, 296.0)
Concentration of dimethyl sulfide (ppb)	10.0 (6.0, 87.0)	3.0 (2.0, 40.0)
Number of teeth with PPD > 4 mm (n)	1.0 (0, 2.0)	0 (0, 1.0)
Number of teeth with BOP (n)	3.0 (0, 4.0)	2.0 (1.0, 4.0)
DI-S	0.17 (0, 0.33)	0 (0, 0.33)
Bacterial counts on the tongue dorsum (×10^5^ CFU/mL)	311 (101, 383)	131 (93.8, 272)

^1^ Median (interquartile range). ^2^ Number (%).

**Table 3 healthcare-12-00980-t003:** Characteristics of participants at 1 week after PDT.

	Intervention Group (N = 11)	Control Group (N = 11)	*p*
Concentration of hydrogen sulfide (ppb)	228.5 (69.0, 339.5)	108.0 (37.0, 228.0)	0.387
Concentration of methyl mercaptan (ppb)	166.0 (31.0, 278.0)	37.0 (7.0, 227.0)	0.171
Concentration of dimethyl sulfide (ppb)	11.0 (1.0, 55.0)	5.0 (3.0, 15.8)	0.654
Number of teeth with PPD > 4 mm (n)	2.0 (0, 2.0)	0 (0, 1.0)	0.270
Number of teeth with BOP (n)	1.0 (0, 2.0)	2.0 (1.0, 4.0)	0.401
DI-S	0.17 (0, 0.33)	0 (0, 0.33)	0.652
Bacterial counts on the tongue dorsum (×10^5^ CFU/mL)	201 (166, 323)	98.9 (28.1, 197)	0.040

Median (interquartile range). Mann–Whitney’s *U* test.

**Table 4 healthcare-12-00980-t004:** Comparison of the rate of change between baseline and 1 week after PDT.

	Intervention Group (N = 11)	Control Group (N = 11)	*p*
Concentration of hydrogen sulfide (%)	−7.2 (−50.6, 63.7)	−14.0 (−72.7, 178.3)	1.000
Concentration of methyl mercaptan (%)	−27.9 (−33.3, 202.2)	16.7 (−73.3, 188.9)	0.606
Concentration of dimethyl sulfide (%)	−21.8 (−88.9, 100.0)	−20.0 (−80.5, 240.6)	0.654
Bacterial counts on the tongue dorsum (%)	−32.9 (−43.3, 65.2)	−27.9 (−50.2, 4.5)	0.606

(One week after PDT − baseline)/baseline × 100. Median (interquartile range). Mann–Whitney’s *U* test.

## Data Availability

No new data were created or analyzed in this study. Data sharing is not applicable to this article.

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
