# Peer review of "Effect of Antimicrobial Photodynamic Therapy on the Tongue Dorsum on Reducing Halitosis and the Duration of the Effect: A Randomized Clinical Trial"

_healthcare, 2024, doi:10.3390/healthcare12100980_

Round 1
Reviewer 1 Report
Comments and Suggestions for Authors
The research is a single blind randomized clinical trial, the CONSORT check list could be verified. Unfortunately, some aspects must be improved:
1. Must explain better the PDT use: it is injected or applied?
2. It is not clear what is the difference between the intervention group and the control group? The description is: “ Participants were recruited from among the students at Okayama University Dental School. Those who complained of oral halitosis and were over 20 years of age were included”.
3. In the results reported, the intervention and control group presented very discrepant initial parameters. What is the author’s explanation for these results?
4. The laser parameters must include at least these information: irradiance (mW/cm2), fluence (J/cm2), Spot size at the probe tip (cm2)
5. Discussion: line 244 to 251- observe that the laser protocol methodologies to be compared requires the description of more parameters be compared than time
Author Response
The research is a single blind randomized clinical trial, the CONSORT check list could be verified. Unfortunately, some aspects must be improved:
1. Must explain better the PDT use: it is injected or applied?
Thank you for your comment. PDT was applied, not injected, on the tongue dorsum. We wrote it on Line 116.
2. It is not clear what is the difference between the intervention group and the control group? The description is: “Participants were recruited from among the students at Okayama University Dental School. Those who complained of oral halitosis and were over 20 years of age were included”.
Thank you for your comment. We have added the procedure in the control group to understand the difference following the reviewer’s suggestion on Line 122.
3. In the results reported, the intervention and control group presented very discrepant initial parameters. What is the author’s explanation for these results?
Thank you for your comment. Discrepancy of parameters at baseline arose because stratification by VSC concentration was not possible at baseline. However, there were no significant differences in all valuables between the group (p > 0.05, Mann-Whitney U test). We compared the rate of change between baseline and one week after PDT for each subject.
4. The laser parameters must include at least these information: irradiance (mW/cm2), fluence (J/cm2), Spot size at the probe tip (cm2)
Thank you for your comment. We have added the information of the laser parameters (Lines 114-115).
5. Discussion: line 244 to 251- observe that the laser protocol methodologies to be compared requires the description of more parameters be compared than time
Thank you for your comment. We have considered not only the irradiation time but also the wavelength and intensity of the laser. We have revised the manuscript (Lines 263-271).
Reviewer 2 Report
Comments and Suggestions for Authors
1. It would be helpful if you included a chart with all the indexes you've measured and what they suggest as background information in the introduction.
2. Maybe generate graphs for statistically significant comparison (which makes them more reader-friendly) while moving these charts (Table 1-3) into supplementary.
Comments on the Quality of English LanguageThe general content is pretty clear, but the flow can be further improved by proofreading done by a professional academic writer.
Author Response
1. It would be helpful if you included a chart with all the indexes you've measured and what they suggest as background information in the introduction.
Thank you for your comment. We have added a new table to show each variable and information following the reviewer’s suggestion (Table 1).
2. Maybe generate graphs for statistically significant comparison (which makes them more reader-friendly) while moving these charts (Table 1-3) into supplementary.
Thank you for your comment. In order to show the median value and distribution, we present as tables rather than a graphs.
Reviewer 3 Report
Comments and Suggestions for Authors
L50 : Antibiotics use in the treatment of (standard) halitosis is not recommended!
As PDT without SRP is not recommended in periodontology, the absence of a mechanical action on the tongue surface, even with PDT, has low or no effect. This study demonstrates and confirms that which is a good point to avoid unuseful overtreatment with new “new PDT” devices.
Comments on the Quality of English Language
Fine
Author Response
L50: Antibiotics use in the treatment of (standard) halitosis is not recommended!
Thank you for your comment. We have changed “antibiotics” to “antimicrobials” following the reviewer’s suggestion on Line 50.
As PDT without SRP is not recommended in periodontology, the absence of a mechanical action on the tongue surface, even with PDT, has low or no effect. This study demonstrates and confirms that which is a good point to avoid unuseful overtreatment with new “new PDT” devices.
Thank you for your comments. As you describe, the results of our study made us think that mechanical action on the tongue surface is necessary to reduce halitosis.
Reviewer 4 Report
Comments and Suggestions for Authors
Dear colleagues!
Issues of a preventive approach to oral diseases and the search for new medicinal methods are always relevant, especially now, when comorbidity accelerates these processes, and the severe epidemiological situation of past years regarding coronavirus infection has increased the negative impact of hypoxia on the oral mucosa and microbiota
The strengths of your study come from a sound design, robust statistical methods, and a sample that is optimal for the evidence base.
In order to recommend transparency of meaning: in section 2.3 you need to add the number of participants in total and in group categories.
It will be useful to add photographs of the treatment and/or recovery process to the results.
The discussion is based on current references that the authors use from the bibliography
Author Response
Dear colleagues!
Issues of a preventive approach to oral diseases and the search for new medicinal methods are always relevant, especially now, when comorbidity accelerates these processes, and the severe epidemiological situation of past years regarding coronavirus infection has increased the negative impact of hypoxia on the oral mucosa and microbiota
The strengths of your study come from a sound design, robust statistical methods, and a sample that is optimal for the evidence base.
In order to recommend transparency of meaning: in section 2.3 you need to add the number of participants in total and in group categories.
Thank you for your comments. We wrote the number of participants in “2.5. Sample size calculation” (Lines 104-110).
It will be useful to add photographs of the treatment and/or recovery process to the results.
Thank you for your comment. We have added the photograph of the treatment (Figure 1).
The discussion is based on current references that the authors use from the bibliography
Thank you for your comment. We have revised the discussion following other reviewers (Line 248-271).
Reviewer 5 Report
Comments and Suggestions for Authors
According to this manuscript, I would like to express my thanks to the authors for their efforts; it needs a minor revision before evaluating the possibility of publication. I would like to pay attention to the following comments:
- The problem question should be properly addressed in the introduction section.
- The null hypothesis should be added at the end of the introduction section and in the discussion.
- What is the importance of this study for clinical? What is the gap in this field of literature?
- The authors should provide summarized data about the other treatment modalities in the halitosis therapies and their short-term outcomes.
- The possible side effects and suspected advantages of PDT should be mentioned in the introduction.
- In the methodology section, the inclusion and exclusion criteria should be mentioned.
- Which statistical test was used to compare these values?
- The discussion should be improved. References to studies in the same area or that are comparable to your work should be made.
- Please clarify the other limitations of this study.
- And clarified the future perspectives.
Author Response
According to this manuscript, I would like to express my thanks to the authors for their efforts; it needs a minor revision before evaluating the possibility of publication. I would like to pay attention to the following comments:
The problem question should be properly addressed in the introduction section.
Thank you for your comment. We wrote it on Lines 71-72 (little is known about the effect of PDT alone on oral halitosis, and the duration of the effect).
The null hypothesis should be added at the end of the introduction section and in the discussion.
Thank you for your comment. We have added the null hypothesis on Lines 72-73 and 214-215 following the reviewer’s suggestion.
What is the importance of this study for clinical? What is the gap in this field of literature?
Thank you for your comment. The importance of this study for clinical is that new approaches for reducing oral halitosis are required. We wrote it on Line 56.
The authors should provide summarized data about the other treatment modalities in the halitosis therapies and their short-term outcomes.
Thank you for your comment. We have added the summarized data about the other studies on Lines 248-256 following the reviewer’s suggestion.
The possible side effects and suspected advantages of PDT should be mentioned in the introduction.
Thank you for your comment. We have revised the manuscript on Lines 60-61 following the reviewer’s suggestion.
In the methodology section, the inclusion and exclusion criteria should be mentioned.
Thank you for your comment. We wrote the criteria on Line 90-93.
Which statistical test was used to compare these values?
Thank you for your comment. The Mann-Whitney U test was used to assess the significance of differences for each parameter between the two groups (Lines 173-174).
The discussion should be improved. References to studies in the same area or that are comparable to your work should be made.
Thank you for your comment. We have revised the discussion on Lines 245-271 following the reviewer’s suggestion.
Please clarify the other limitations of this study.
Thank you for your comment. We have added other limitations following the reviewer’s suggestion (Lines 280-281).
And clarified the future perspectives.
Thank you for your comment. Increasing the number of PDT sessions may be effective against halitosis. We have added on Lines 225-226.